# Hydroacoustic System in a Biomimetic Underwater Vehicle to Avoid Collision with Vessels with Low-Speed Propellers in a Controlled Environment

**DOI:** 10.3390/s20040968

**Published:** 2020-02-11

**Authors:** Pawel Piskur, Piotr Szymak, Krzysztof Jaskólski, Leszek Flis, Marek Gąsiorowski

**Affiliations:** Polish Naval Academy, Smidowicza St, 69, 81-127 Gdynia, Poland; p.szymak@amw.gdynia.pl (P.S.); k.jaskolski@amw.gdynia.pl (K.J.); l.flis@amw.gdynia.pl (L.F.); m.gasiorowski@amw.gdynia.pl (M.G.)

**Keywords:** digital signal processing, hydrophone, passive detection of obstacles, Generalized Cross-Correlation (GCC), Biomimetic Underwater Vehicle (BUV), Autonomous Underwater Vehicle (AUV), direction of arrival (DOA)

## Abstract

In this paper, a hydroacoustic system designed for a biomimetic underwater vehicle (BUV) is presented. The Biomimetic Underwater Vehicle No. 2 (BUV2) is a next-generation BUV built within the ambit of SABUVIS, a European Defense Agency project (category B). Our main efforts were devoted to designing the system so that it will avoid collisions with vessels with low-speed propellers, e.g., submarines. Verification measurements were taken in a lake using a propeller-driven pontoon with a spectrum similar to that produced by a submarine propulsion system. Here, we describe the hydroacoustic signal used, with careful consideration of the filter and method of estimation for the bearings of the moving obstacle. Two algorithms for passive obstacle detection were used, and the results are discussed herein.

## 1. Introduction

In an underwater environment, hydrophones can be widely used for seismic exploration [1], tracking the movement of marine mammals [2], the directional angle estimation of underwater acoustic sources [3], communications purposes [4,5], and the autonomous navigation of underwater vehicles [6], etc. 

A hydroacoustic system can be stationary or mobile. In a stationary system, there are no restrictions on the distance between hydrophones, the number of hydrophones, memory size, or the system’s dimensions and mass in general. The preliminary results from array experiments of six towed hydrophones showed sound propagation and attenuation [7], where the tow cable was 10 m long with neutral buoyancy. Signals were recorded in a 44,100 samples/s format with uncompressed 16-bit resolution. Six hydrophone groups were spaced 0.75 m apart (λ/2 for 1 kHz). Moreover, the acoustic source transmitted signals in different frequency bands of 25–40 kHz. However, in this designed system, the acoustic spectrum was in the range below 1 kHz, which had a strong impact on the desired distance between the hydrophones and the implemented method for collision detection. 

Although it is more convenient to analyze signals from an array with a large number of sensors distributed over a wide space, during the realization of the SABUVIS project, an assumption was made about the total number of hydrophones used. First, it was assumed that the biomimetic underwater vehicle (BUV) hull had to be compact, with no protruding parts. The number of hydrophones mounted on the BUV was reduced to one pair due to the short length of the BUV (and the rising costs in scenarios with multiple BUVs).

A stereo hydrophone system mounted on an Autonomous Underwater Vehicle (AUV) was presented in Reference [8], and it was able to estimate the position of a moving acoustic tag, while remaining at a distance. However, the hydrophone rig was attached to the bottom of the AUV, with each hydrophone suspended 0.4 m below the AUV and 2.4 m apart. Moreover, the signal frequency was much higher than that which can be produced by vessels with low-speed propellers, e.g., submarines.

In the SABUVIS project, the passive measurement system was based on two hydrophones mounted on the upper part of the BUV No. 2 (BUV2) 1.25 m apart. Furthermore, the measured signal had a low frequency that is typical of submarine propellers. The frequency of a typical hydroacoustic wave from a submarine propeller is about 20 Hz, and the length of the wave is λ = 75 m, with a sound velocity of approximately 1500 m/s. To calculate the distance between hydrophones requires an adequate sampling frequency and the length of the measuring signal used for signal processing. When the signals are sampled at a high frequency, copious amounts of data are gathered and must be stored or transmitted for processing; thus, high-speed communication is required. The precision of the bearing estimation depends on the number of zone divisions and the sampling resolution.

The goal of a hydroacoustic system is uninterrupted collision avoidance, which is why recorded signals cannot be analyzed offline, as they were for the sperm whale tracking system presented in Reference [2], where recorded data were saved and archived in a memory card and then analyzed after the completion of the mission. With the BUV2, measurements must be made in real time, and in the case of collision detection, the course and/or depth of the vehicle must change.

In Reference [9], a novel approach using a Hilbert-Huang transform and a uniform linear array of hydrophones was proposed for the joint bearings and range estimation of multiple targets. The proposed method (for target location estimation) had a simple algorithm derived from the transformation of a very complicated nonlinear estimation problem. Unfortunately, the results and discussion only included a noise-free signal, which is impossible to obtain in a real environment.

In Reference [10], self-adjusting characteristics for an adaptive line enhancer for a time domain broadband beamformer were introduced, and then a self-steering broadband beamformer was proposed. However, this set-up requires prior information about the target direction, which is unknown in real conditions. Moreover, the algorithm verification was performed only for simulation signals.

A novel direction-of-arrival estimation algorithm for a coprime linear array (CLA) based on the multiple invariance estimation of signal parameters using rotational invariance techniques (MI-ESPIRT) and a lookup table (LUT) was proposed in Reference [11]. In another paper [12], the assumption was made that a vehicle would be equipped with two hydrophones and that the acoustic source would transmit a specific signal repeatedly. The prediction probability was updated using a generalized cross-correlation function with a verification process that included entropy measurements. The effectiveness of a method based on a MUSIC algorithm has also been verified and compared to other methods created from computer simulation results [13]. Reference [14] dealt with a robust direction-of-arrival (DOA) estimation method for sources with known waveforms undergoing Doppler shifts. That moving obstacle detection system used a modified algorithm for DOA estimation. In practice, it was difficult to evaluate an unknown number of signals impinging on an array simultaneously, each from an unknown direction and with an unknown amplitude. In addition, the received signals were always corrupted by noise.

Nevertheless, there are several available methods for estimating the number of signals and their direction: most of them require assumptions about the known number of signals, and in addition, the number of signals has to be lower than the number of sensors (or the number of sensors has to be higher than one pair).

We considered the idea of one pair of hydrophones for the detection of moving obstacles, and an algorithm for a low-frequency signal was proposed and then experimentally verified. In an emergency situation where the BUV2 is in a crowded zone, the buoyancy can be changed using the ballast tank and the BUV2 can avoid collision by ascending or submerging. The approach to a ship can be estimated using a sound pressure drop in a 20 * log (range) in an acoustic-free field. In addition, the Doppler effect can be included due to the frequency changing if the vessel is moving at a high linear velocity and the rotational velocity of the impeller is low.

The results of the research presented in this paper are a continuation of the results described in Reference [15]. The simulation results from that research, which used an algorithm for the passive detection of moving obstacles, were presented in Reference [16]. To measure and verify the system calibration, laboratory tests with undistorted sinusoidal signals with known frequencies, bearings, and distances were performed in a swimming pool and in a lake and were described in Reference [17]. Here, the results of the final system verification are presented.

The research questions this paper attempts to answer are as follows:Is it possible to design a passive anticollision system using only two hydrophones mounted on the compact hull of a BUV?Is it possible to design a computationally undemanding method of estimating the DOA of a vessel with low-speed propellers for implementation in a small AUV?

## 2. Materials and Methods

### 2.1. Description of a Hydroacoustic System

The hydroacoustic system was realized using HTI-96-MIN hydrophones with omnidirectional characteristics and a sensitivity of −165 dB re: 1 V/uPa. The hydrophones were mounted on the upper part of the BUV2, 1.25 m apart (Figure 1). Given that the speed of sound in water is (approximately) equal to 1500 m/s, an acoustic wave needs less than one millisecond (about 0.83 ms) to move between the two hydrophones if the source of the sound is located along the BUV2 axial symmetry. For an acoustic signal with a frequency equal to 20 Hz (typical for submarine propellers), the hydroacoustic wavelength is equal to 75 m and the wave period is 0.05 s. The sampling frequency of an ADC should take into account the distance between hydrophones, as well as the length of a hydroacoustic wave. For hydroacoustic signals with low frequencies (for example, in a warship propeller [18]), the length of a hydroacoustic wave can be 100 m long, which is why data sampled for a long time have to be gathered for course calculation. The sampling frequency depends on the accuracy of the course estimation. Here, the sampling frequency used during the tests was the minimum accepted sampling frequency (equal to 10 kHz), where the bearing could be divided into approximately 10-degree zones. Additionally, an analog amplifier (with a gain of 32 dB) was added before analog-to-digital conversion. The hydrophone signals were converted from analog to digital form with a 12-bit sampling resolution. Then, the data were analyzed in a microcontroller unit (MCU), where filtration and bearing estimation were realized.

In real-time applications, rapid computation (e.g., with digital signal processors) is essential. Taking into consideration hydrophone sensitivity, analog-to-digital conversion, the accuracy of the measuring path, and the disturbances in real environments, a microcontroller called a TI C2000 with a DSP TMS320F28379D [19] was used during the preliminary tests. Then, due to the risk of heap overwrite, an NI sbRio-9603 was used (with the same sampling frequency of 10 kHz).

### 2.2. Measurement Site

The measurements were performed in a lake with a depth of up to 6 m. It was assumed that the signals from the vessel’s propellers had a constant rotational speed. The rotational speed was approximately equal to a submarine propeller; however, due to logistical restrictions, a pontoon (a ‘‘Kolibri KM-330PP’’) with an electrically powered propeller (Minn Kota) was used as a source of underwater sound. The pontoon was directed via a manually operated remote control, and as it approached and moved away, its signals were measured using both hydrophones. In Figure 2, the time and frequency domains of the measured signals are depicted.

### 2.3. Short-Time Fourier Transform (STFT)

Signals in the time domain obfuscate the frequency, so even if the frequency is known, the time of appearance is unknown. Instead of looking at the Digital Fourier Transform (DFT) of an entire signal in one step, the entire signal was divided into frames of length *L*. Then, the DFT of each frame was calculated. To reduce artifacts at the boundary, every frame overlapped. The overlap between adjoining frames was equal to half of the frame (L/2 samples). The DFT coefficient was indexed by two variables, *n* and *k*.

Each analyzed frame started at point *n* and had a length *L* according to the equation
(1)$$X\left[n,k\right]=\overset{L-1}{\underset{m=0}{\sum }}X\left[n+m\right]w\left[m\right]e^{-jk\frac{2\pi }{N}m}$$
where *m* is the starting point for the localized DFT; *k* is the DFT index for every frame; *m* is the reference time; *L* is the size of the DFT window; w is a Hamming window; and *N* is the total number of gathered signal samples.

For the Short-Time Fourier Transform (STFT) computations, both signals were divided into 10,000 samples and given a Hamming window, with 5000 samples of overlap between adjoining segments. Each frame was Fourier-transformed, and the complex result was added to a matrix, which recorded the magnitude and phase for each point in terms of time and frequency. The Short-Time Fourier Transform was calculated for every X[n], X[n+1], X[n+2], …, X[n+L−1], and the magnitudes of the square of $|X\left[n,k\right]{|}^{2}$ are presented as spectrograms in Figure 3 and Figure 4 (the pontoon moving away from the BUV and approaching the BUV, respectively). The spectral slices were placed subsequently to each other to obtain an image-like depiction of the time-varying spectrum. A spectrogram is a clever way of showing this time-varying spectral information in a single plot.

For a sample frequency of Fs = 10 kHz, the highest positive frequency was ½Fs (equal to 5 kHz). The frequency resolution was Fs/L Hz, and the width of the time slices was L*Ts s, where Ts = 1/Fs. Thus, the maximum frequency was 5 kHz, but the measured signal from the submarine propeller was below 100 Hz, which is why the spectrogram was magnified in a range from 0 to 100 Hz.

The position and type of window as well as the window coefficients influence the amount of energy that is captured in STFTs. With short windows, there is very good localization in time, but the frequency bands are extremely wide. In addition, a very short window creates artifacts in a high-frequency range. Long windows have more DFT points and a higher frequency resolution, which is why more signal components appear: as a result, the system is less precise in terms of time. On the other hand, short windows have a wideband spectrogram, and the transition is much more precise, but short windows give fewer DFT points, and the frequency resolution is poor. We assumed that the type of window would not have a significant impact on the DFT due to the 20-fold higher length of the window in comparison to the period of the measured signal.

In Figure 3, some regions are marked: the first region comes from the BUV propulsion system and could be attenuated by a high-pass filter. The second region is from a real signal from the propeller, with higher harmonics marked as region no. 3. The fourth region, which has a constant frequency, is noise from an unknown source.

It can be seen in Figure 3 that from the beginning of the measurements, the signals were about 18–19 Hz (frequency). Due to the Doppler effect, the frequency decreased, as did the amplitude of the signal, while the pontoon was moving away. The second harmonic was 36–38 Hz, with a proportionally lower amplitude. There was also a signal with a frequency below 40 Hz for 10–22 s during the measurements, but the signal had a constant frequency, and the source of the signal is unknown.

At the beginning of the pontoon’s movements, when the pontoon was starting from zero to a nominal pontoon velocity (Figure 4), there was a signal for the first 5 s, but the acoustic wave was not strong enough to be measured by the hydrophone system until it was 12 s away. The signal frequency was constant until there was only a short distance between the pontoon and the BUV in the last stage of measurement.

The frequency changes according to the Doppler effect are
(2)$$f_{O}=f_{P}\frac{v\pm v_{BUV}}{v\pm v_{P}}$$
where $f_{O}$ is the observed frequency; $f_{P}$ is the frequency emitted by the propeller; $v_{P}$ is the linear velocity of the source of the sound (here, a pontoon with a propeller powered by an electric motor); $v_{BUV}$ is the velocity of the BUV; and v is the sound velocity.

The sign ± depends on the direction of movement between the BUV and the source of the sound (here, a pontoon).

Taking Equation (2) into consideration, it is clear that the Doppler effect is strongly related to the velocity of the source of sound, $v_{BUV}$. This can also be observed in the spectrograms (Figure 3 and Figure 4). At the beginning of both the approaches and departures, the velocity $v_{BUV}$ was small, because the BUV started its motion from zero velocity. Therefore, a bigger Doppler effect, i.e., bigger changes in the observed frequency $f_{O}$, could be seen at the end of both tests.

There was also a second-order harmonic visible in the measured signals (Figure 3 and Figure 4), but it was much more difficult to distinguish from noise due to the lower power of this harmonic. The low-frequency noise mainly came from waves and the self-propulsion system of the BUV. Although low-frequency noise can be removed with a low-band filter, distinguishing the propeller hydroacoustic signal from noise is not an easy task. For that reason, a signal-to-noise ratio was calculated and is discussed in the next section.

### 2.4. Signal-to-Noise Ratio

To distinguish signals from noise, a signal-to-noise ratio (SNR) method was used to quantify the distortion and noise. The SNR analysis was performed for every second of measurement. The spectrogram created from the SNR analysis included all nonfundamental spectral components in the Nyquist frequency range with constant components, fundamentals, and harmonics. To determine the SNR, the following formula was used:(3)$$SNR=20\log\left(\frac{S}{N}\right)\;[\mathrm{dB}]$$
where S is the sum of the squared magnitudes of the signal, and N is the sum of the squared magnitudes of the noise.

In Figure 5 and Figure 6, it can be seen that any elements with values smaller than SNR = 80 (dB) were set at zero. In Figure 5, the ratio of the signal-to-noise spectrum is presented for when the pontoon was moving away from the BUV.

In Figure 6, the spectrum of the SNR analysis for when the pontoon was approaching the BUV is presented. In addition to the propeller signal, the constant components as well as a low-frequency distortion signal can be observed. The signal’s constant components and the low-frequency signal will be removed in the next step of analysis.

If the main harmonic frequency is known, then one of the higher harmonic frequencies can be considered if the beam estimation is provided with a higher resolution. Higher harmonic frequencies were limited due to the dumping of higher frequencies in water. Here, the main frequency component was used to estimate the course of the pontoon relative to the BUV.

### 2.5. Signal Filtration

The principle of measuring the offset between two signals is the main condition for estimating a course. To distinguish the signal of the propeller and to eliminate phase distortion, a finite impulse response (FIR) with a linear phase filter was designed and implemented [20]. A zero-phase filter was needed to keep the phase of both signals undisturbed. After filtering the data in a forward direction, the filtered sequence was reversed and run back through the filter. As a result of the filtration process, the signals should have had no phase distortion.

A passband linear phase filter was implemented in the algorithm with the following parameters (Figure 7): f(stop1)=15 Hz; f(pass1)=16 Hz; f(pass2)=19 Hz; and f(stop2)=20 Hz.

The first two parameters for the acoustic signal of the BUV self-propeller system were determined, and the following parameters were assumed for the maximal range of the expected frequency of the propeller. To find the signal of the propeller, the largest nonzero spectral component was searched for. To find the fundamental frequency, a Kaiser window with ample sidelobe attenuation was used. The constant component of the signal was excluded from the calculation.

According to the filter model’s assumptions, the bandwidth between the first passband frequency and the second passband frequency will be passed, and other frequencies will attenuate. The lower band of the filter was established to reduce the noise from the BUV self-propeller system (decreasing the impact of the compact hull), while the upper band was intended to reduce the number of higher harmonics in the SNR analysis due to the fact that higher frequencies are attenuated in water environments.

It can be seen that the amplitude decreased over time as the pontoon moved away from the BUV (Figure 8a), and the amplitude increased over time as the pontoon approached the BUV (Figure 8b).

### 2.6. The DOA Estimation Method

The main idea for the bearing estimation of an underwater source of sound is to find the delay between signals (measured by the two hydrophones). The time delay between two signals depends on the shift between the samples ∆*n* and the sampling frequency *f_s_*. The maximal resolution of this method is equal to the sampling period *T_s_ = 1/f_s_*. By interchanging the roles of $h_{1}$ and $h_{2}$ and recomputing the cross-correlation sequence, the value of $R_{h_{1},h_{2}}\left(∆n\right)$ can be obtained for negative shifts.

The cross-correlation sequence between two hydroacoustic signals (h_1_ and h_2_) represents the lag between the two signals. This maximum corresponds to the sample shift ∆n with the greatest similarity between signals and is defined as
(4)$$R_{h_{1},h_{2}}\left(∆n\right)=\overset{n}{\underset{i=1}{\sum }}h_{1}\left(i\right)*h_{2}\left(i\pm i*∆n\right)$$
where *n* is the number of gathered samples.

A positive or negative shift determines the area in front of or behind the underwater vehicle. Because of the axial symmetry of the BUV, when a course collision is detected, the BUV must change course to verify if the source of sound is from the left or from the right side of the BUV axial symmetry.

To improve the cross-correlation function, both signals were normalized, and the average value of both signals was then shifted to zero according to Equation (5):(5)$$Rn_{h_{1},h_{2}}\left(∆n\right)=\overset{n}{\underset{i=1}{\sum }}(\frac{h_{1}\left(i\right)}{abs\left(\max\left(h_{1}\left(i\right)\right)\right)}-mean\left(h_{1}\left(i\right)\right)*\left(\frac{(h_{2}\left(i\pm i*∆n\right)}{abs\left(\max\left(h_{2}\left(i\pm i*∆n\right)\right)\right)}-mean\left(h_{2}\left(i\right)\right)\right)$$

Taking into consideration the specific conditions of measurements, a new method was proposed and verified. Because of the length of the data vector and the power-consuming calculation needed for the cross-correlation algorithm, the new method leads to the discovery of the lag between both measured signals by determining the offset and calculating the zero-point intersection. There is no need to gather thousands of datapoints (only several dozen). If one signal crosses the zero point, the next signal is checked at the neighborhood point when the zero-point crossing appears. According to hydroacoustic system specification theory, the shift cannot be greater than the number of samples gathered, with restrictions on the distance between hydrophones and the velocity of sound in water. More detailed information is included in Reference [15].

This method was invented because of the technical conditions of the measurements: instead of requiring power-consuming calculations for the cross-correlation algorithm, the method takes into account that both signals are long and have thousands of samples (but the lag does range from zero to nine samples). The second algorithm compares the nearest two cross-zero points and calculates the distance between them.

Since the distance between the hydrophones is 1.25 m and the soundwave length is 75 m, no more than 1.5% of samples are swift during the correlation calculations. This significantly reduces the memory and computational power demand.

This new method (called “cross-zero”) is a modification of the one presented in Reference [15]. The two signals were checked simultaneously with additional conditions: the nearest cross-zero point should be on a rising slope according to Equation (6), which is
(6)$$h_{1}\left(i\right)\leq 0\;and\;h_{1}\left(i+1\right)\geq 0\;h_{2}\left(j\right)\leq 0\;and\;h_{2}\left(j+1\right)\geq 0$$
where *i* and *j* are the numbers representing the next *h_1_* and *h_2_* signal samples, respectively.

Here, the method of measurement is provided for the first signal after filtration, and if the zero-crossing point is detected, then the second signal is analyzed within the enclosed area. This means only the range between a -10 and 10 sample offset is considered.

## 3. Results

Measurements were done using a pontoon both approaching and departing, with four different courses: 0 [deg], 30 [deg], 60 [deg], and 90 [deg]. The course was changed using the BUV’s course stabilization system, while the pontoon followed the trajectory depicted in Figure 9. The 0 [deg] notation is for the pontoon on a collision course—30 [deg] represents the border of the collision course, and 60 [deg] and 90 [deg] mean that the pontoon is moving to a safe zone. At 90 [deg], the time shift between both signals was equal to zero due to the pontoon being positioned perpendicularly to the BUV2 axial symmetry. The pontoon moved with a maximal velocity equal to 4 m/s.

In Figure 10, Figure 11, Figure 12 and Figure 13, the experimental results for the different course approximations estimated using the two methods are presented. In Figure 10, the results from the experimental verification of the GCC method are presented (a pontoon approaching the BUV2). When the pontoon started, the signal was strong enough to be measured by the hydrophones. After 6–7 s, the signal was too weak to be analyzed (until the pontoon came closer to the BUV2 at 10–12 s). For the next 20 s of measurement, the course was correctly determined.

In Figure 11, the results of the experimental verification of the GCC method are presented (for a pontoon moving away from the BUV2, taking four different courses). The bearing estimation was properly estimated until there was only a short distance between the pontoon and the BUV2. After 10 s of increasing distance, the accuracy of the beam estimation decreased.

Because the GCC method is power- and memory-consuming, the cross-zero method was implemented and verified using the same measurement conditions.

In Figure 12, the results of the cross-zero method verification are presented for a pontoon approaching the BUV2. As with the GCC results, the hydroacoustic beam was estimated with better accuracy as the pontoon started than it was a few seconds later, when the pontoon was moving at a constant velocity but the distance was too far to effectively measure the signal. After 10 s of movement, the hydroacoustic system was able to measure a signal strong enough to set the course correctly.

In Figure 13, the results of the experimental verification of the cross-zero method are presented (for a pontoon moving away from the BUV2, taking four different courses). The beam estimation results were similar to the results achieved with the GCC method. The course was estimated properly until the distance between the pontoon and the BUV2 was short enough to be measured with the hydrophones.

The course error estimation was not calculated in this paper due to many factors having an impact on accuracy. During the test in the real environment, it was observed that the measurement accuracy depended on the source of the signal, which means that with a more powerful propeller, the accuracy would increase.

The main conclusion is that the presented hydroacoustic system can detect the bearings of moving obstacles. The second result worth mentioning is that the proposed cross-zero algorithm is much less power- and memory-consuming in comparison to the cross-correlation method. This feature is important for autonomous vehicles because their operating times depend on the electric energy consumed.

## 4. Discussion

The experimental results confirmed that it is possible to design and implement a system for detecting obstacles based only on two hydrophones in a BUV. The hydroacoustic measurement methods are described here in detail to allow others to replicate and use them. We make the following suggestions:(1)Hydrophone sensitivity and the gain of the amplifier module should be selected appropriately based on the expected distance of detection;(2)The number of hydrophones in the array is significant. If only two sensors are used, the broadside angle of arrival can only be estimated;(3)The distance between hydrophones, the desired zone resolution, the velocity of sound, and the sampling frequency should be calculated;(4)The ADC resolution and the range of signals should be taken into consideration;(5)Because most of the filters introduce a variable phase shift into the filtered signals, FIR filters with a constant phase shift should be implemented;(6)The bearing estimation method should be appropriate according to the chosen hydrophone array and target platform (stationary, mobile, small, large, etc.);(7)Similar accuracy can be achieved using either the cross-correlation or cross-zero method, i.e., either a more or less complicated method; and(8)The proposed cross-zero method has sufficient accuracy and is less computationally demanding.

From these conclusions, it is clear that the bearing estimation of sources of underwater sound, even at low frequency, can be performed in a controlled environment. Therefore, further research on this type of bearing estimation—using only two hydrophones—is justified and advisable. The activity could be directed, e.g., the impact of the signal filter range on the accuracy of the bearing estimation or tests with hydrophones with higher sensitivity.

The presented devices and methods for analyzing underwater signals will be further developed in an underwater communications system [21] that has two hydrophones employed as an acoustic transmitter and receiver.

## Figures and Tables

**Figure 1 sensors-20-00968-f001:**
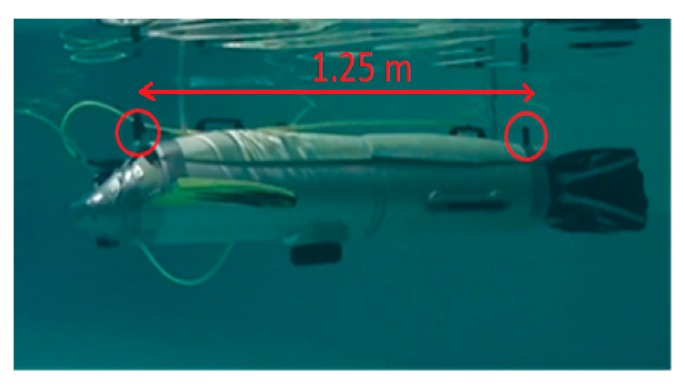
The Biomimetic Underwater Vehicle No. 2 (BUV2) is shown here underwater, with the hydrophones marked by red circles.

**Figure 2 sensors-20-00968-f002:**
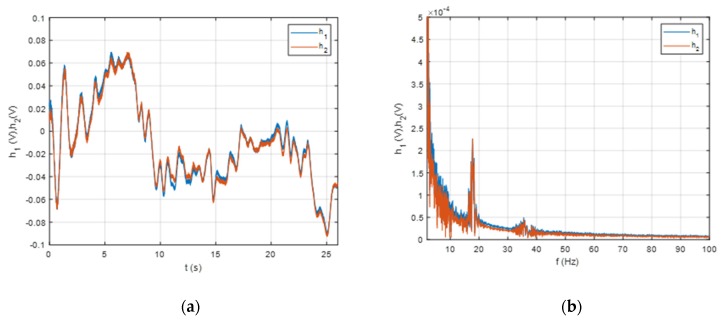
Measured signals from the two hydrophones (h1 and h2): (**a**) the time domain; (**b**) the frequency domain.

**Figure 3 sensors-20-00968-f003:**
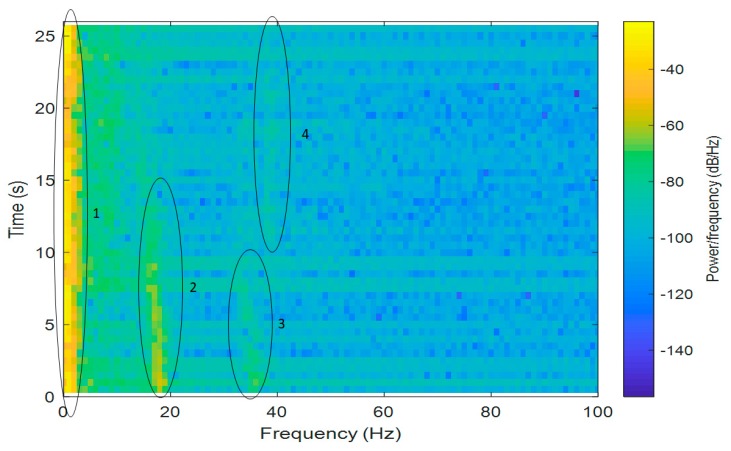
A spectrogram of measured hydroacoustic signals as the pontoon moved away from the BUV.

**Figure 4 sensors-20-00968-f004:**
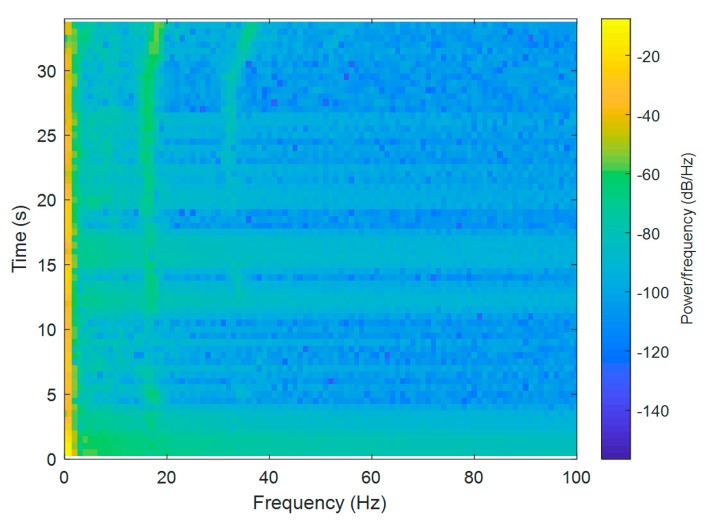
A spectrogram of the hydroacoustic signal measured during the approach of the BUV.

**Figure 5 sensors-20-00968-f005:**
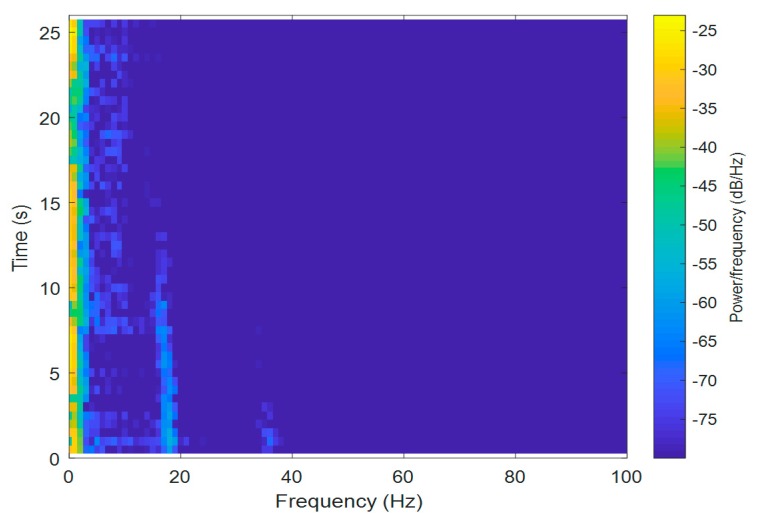
Spectrum after a signal-to-noise ratio (SNR) analysis for the pontoon moving away from the BUV.

**Figure 6 sensors-20-00968-f006:**
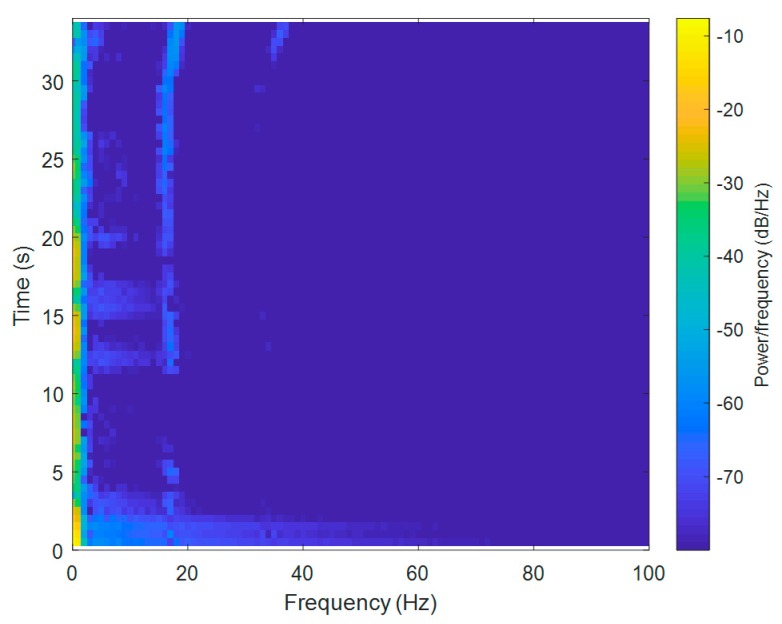
Spectrum after an SNR analysis for the pontoon approaching the BUV.

**Figure 7 sensors-20-00968-f007:**
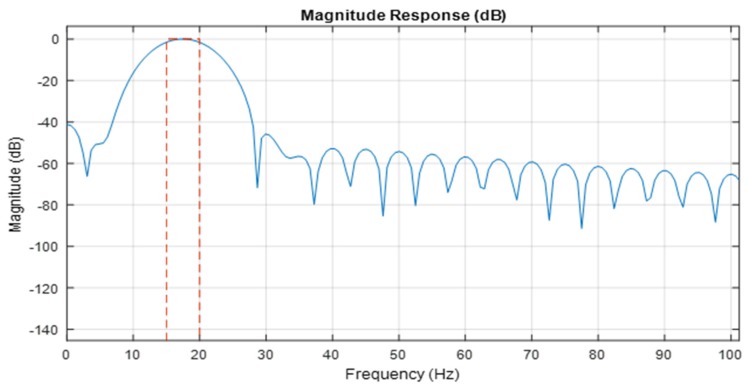
The passband linear phase filter.

**Figure 8 sensors-20-00968-f008:**
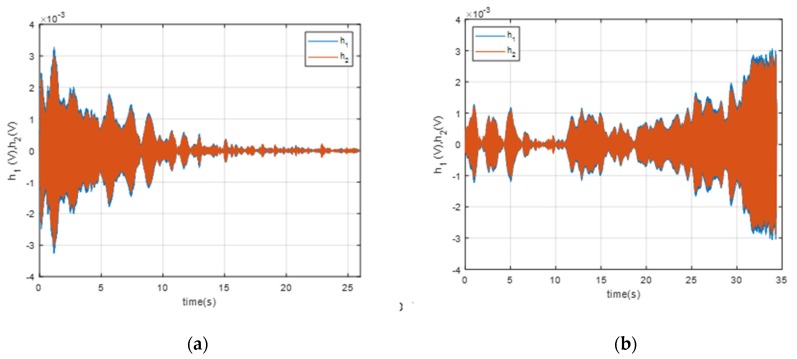
The signals measured after filtration with the passband linear phase filter in terms of the source of the sound: (**a**) approaching the BUV; (**b**) departing the BUV.

**Figure 9 sensors-20-00968-f009:**
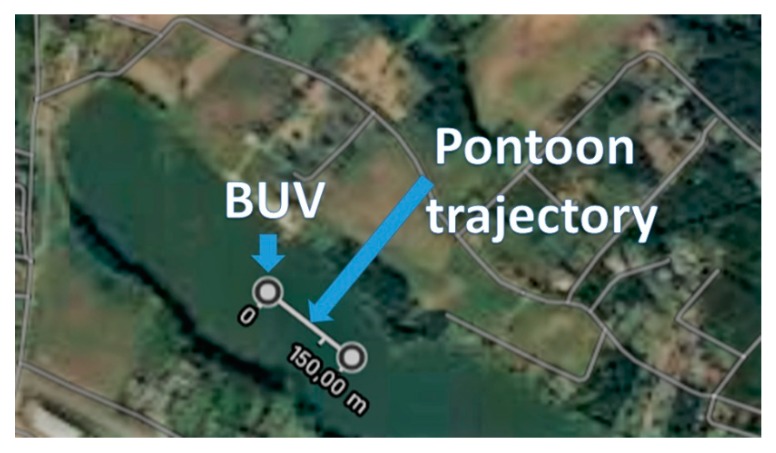
The pontoon trajectory relative to the BUV.

**Figure 10 sensors-20-00968-f010:**
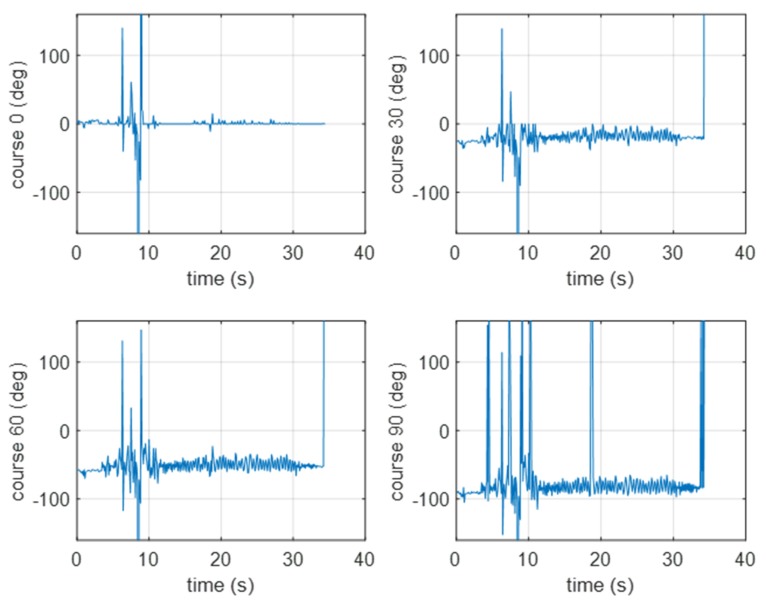
Results of the bearing estimation using the cross-correlation method (for a pontoon approaching the BUV2 at 0, 30, 60, or 90 [deg]).

**Figure 11 sensors-20-00968-f011:**
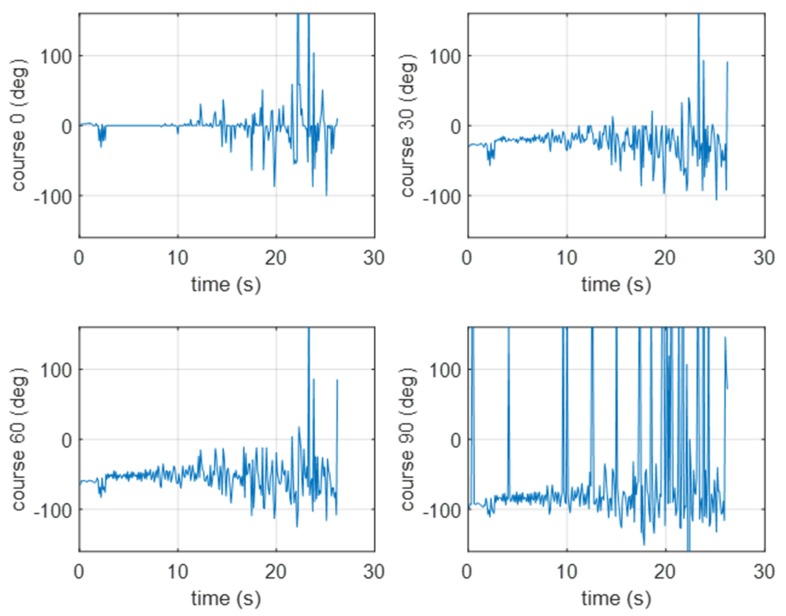
Results of the bearing estimation using the cross-correlation method (for a pontoon moving away from the BUV2 at 0, 30, 60, or 90 [deg]).

**Figure 12 sensors-20-00968-f012:**
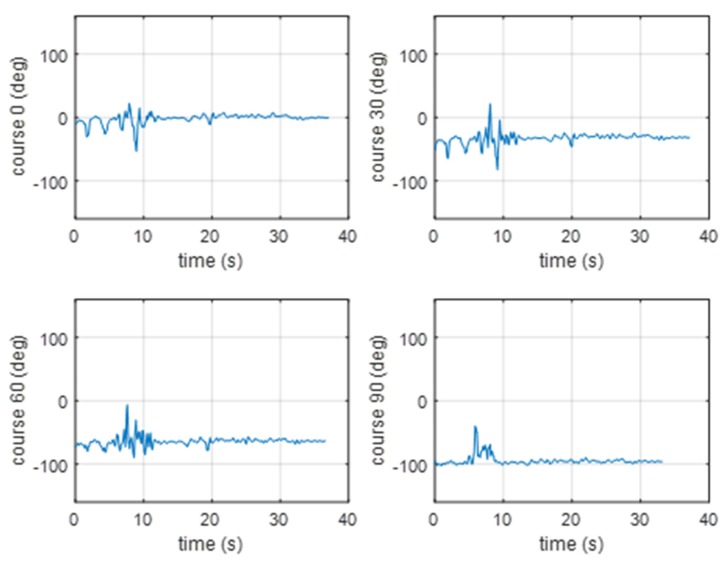
Results of the bearing estimation using the cross-zero method (for a pontoon approaching the BUV2 at 0, 30, 60, or 90 [deg]).

**Figure 13 sensors-20-00968-f013:**
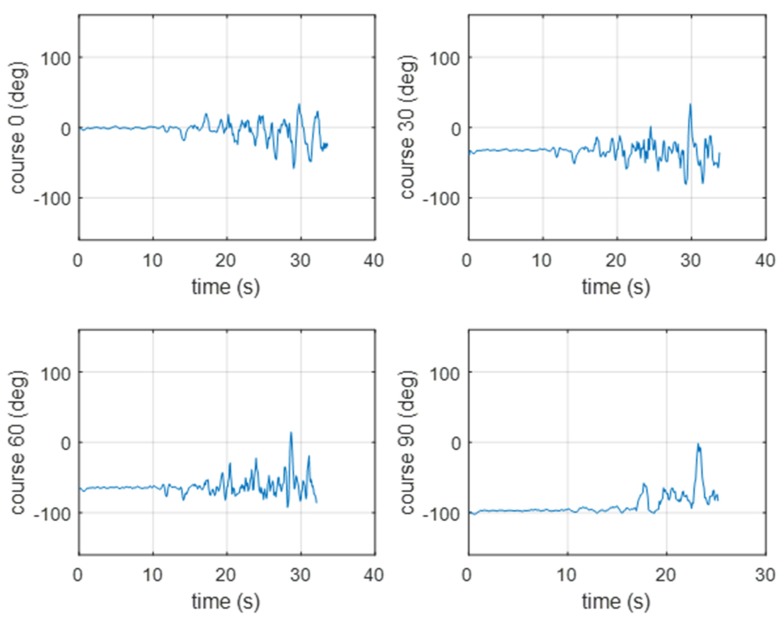
Results of the bearing estimation using the cross-zero method (for a pontoon moving away from the BUV2 at 0, 30, 60, or 90 [deg]).

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
