# Peer review of "Hydroacoustic System in a Biomimetic Underwater Vehicle to Avoid Collision with Vessels with Low-Speed Propellers in a Controlled Environment"

_sensors, 2020, doi:10.3390/s20040968_

Round 1

Reviewer 1 Report

It would be interesting to make some graph of the distance between the pontoon and the vehicle with respect to time. This would allow to calculate the maneuver of the vehicle to avoid the collision

To assess the goodness of the method it would be interesting to know the estimation of the error, even for this type of propeller.

Author Response

Thank you for your review of our paper. All your suggestions have been introduced into revised manuscript. We have answered each of your points below.

“It would be interesting to make some graph of the distance between the pontoon and the vehicle with respect to time. This would allow to calculate the maneuver of the vehicle to avoid the collision”

The additional figure containing the distance between the pontoon and the vehicle has been inserted. Moreover, the explanation in the manuscript text about the velocity and maneuver capabilities of the vehicle has been added.

“To assess the goodness of the method it would be interesting to know the estimation of the error, even for this type of propeller.”

Although many measurements were done only sixteen results (eight data for four different courses) were presented and discussed in the paper.  

Indeed, we agree with you that accuracy analysis provides important information about the system and is always worth including in the technical documentation. But here, according to the authors experiences, there are too many following factors influencing on hydroacoustic propagation that it is problem to make reliable estimation of error in the article:

- the environmental factors (water temperature, water depth, type of bottom, distance from the shore, etc.),

- the measurements factors (type of propeller, depth of the BUV, etc.)

- devices parameters (kind of hydrophones, gain of the amplifier module, ADC resolution, etc.)

- algorithm used parameters (filter range, number of the measured data, etc.)

What is more, the pontoon was not equipped with a specialized gyroscope for course measurement, so the accuracy analysis would not be comprehensive.

The paper mentions that the test are going to be carried out for various range of filters and the accuracy of direct estimation method is to be verified for constant environment condition.

But to be honest, we can't provide enough comprehensive error analysis at this time.

Here, we focused on the difficult task of estimating the course of low speed propellers (as in submarines) using only two hydrophones mounted at a closed distance. All stages of this analysis have been presented. What's more, the new method has been proposed and experimentally verified. During the next research we would like to compare this method with others including statistical error analysis. These tests demands other water basin and development of our measurement stand. Finally, the tests with submarine are expected but probably their results will be unclassified.

Reviewer 2 Report

The paper presents the results of field experiments with an AUV equipped with a hydroacoustic system of two hydrophones. The particular attention is devoted to the signal direction of arrival estimation.

Though the presented results are interesting for the practice and maritime applications, the answer to the question posed by the authors (the possibility of the obstacle detection system with two hydrophones) can not be called persuasive. The issues are:

the significant details of the experimental setting, such as the distances between the AUV and the submarine surrogate, their speeds, attitudes, and other trajectory properties are omitted; for a successful obstacle detection system, the distance to the obstacle is even more crucial than the direction, but this issue is also kept off the consideration; the DOA estimations are only made for constant angles. It is not clear how the algorithms would work for the moving objects when the DOA changes in time.

The mentioned concerns, nevertheless, do not diminish the value of the obtained field data. But I believe that the presentation (starting from the title) should be adjusted either by changing the motivation or by providing more information on the experimental settings and much more discussion on how the results obtained can be used for the moving obstacle detection and collision avoidance.

The reference on the yet unpublished work [17] is disappointing.

Author Response

Thank you for your review of our paper. All your suggestions have been introduced into revised manuscript. We have answered each of your points below.

“…the significant details of the experimental setting, such as the distances between the AUV and the submarine surrogate, their speeds, attitudes, and other trajectory properties are omitted”

Information about distances between the AUV and pontoon imitating submarine, speeds and trajectory has been added to the article.

“…the distance to the obstacle is even more crucial than the direction…”

We agree that the distance is crucial in avoiding collision aspects – that is why all the measurements were done for approaching and moving away. All the experiments were made for the constant direct of arrival but the distance between BUV and pontoon was changed. 

“It is not clear how the algorithms would work for the moving objects when the DOA changes in time.”

Because of the luck of the precision course measurement on board of the pontoon the analysis was not performed for the measurements when the pontoon was moving around the BUV.

Moreover, due to the lake dimensions and weather condition, it was difficult to control precisely the speed and the position on the fixed radius of the pontoon trajectory, which swam around the BUV.

“The mentioned concerns, nevertheless, do not diminish the value of the obtained field data. But I believe that the presentation (starting from the title) should be adjusted either by changing the motivation or by providing more information on the experimental settings and much more discussion on how the results obtained can be used for the moving obstacle detection and collision avoidance.”

The introduction and conclusion were reorganized in more logical order. The motivation was expressed in different way based on suggestions of reviewers.

Moreover, the paper was polishing by native speakers from an MDPI journal.

“The reference on the yet unpublished work [17] is disappointing.”

We apologise for that, but we did not noticed that the publication is on-line first, it means it is not available in databased yet. I hope that this will be done soon. Below please find the information about the paper:

https://hrcak.srce.hr/index.php?show=clanak&id_clanak_jezik=338247&lang=en

Reviewer 3 Report

The paper deals with the description of an hydroacoustic system to avoid collision with vessels with low speed propeller based on a pair of hydrophones part of the equipment of the second generation of Biomimetic Underwater Vehicle developed within the SABUVIS project.
The paper presents the description of the system, a section about the processing of the acquired data based on STFT, S/N ratio, filtering and estimation of direction of arrival and a section with the results based on a single test carried out in a lake using a pontoon to simulate vessel with low speed propellers.
Discussion session is just a collection of recommendation to the reader who wants to replicate and use the methods, but no discussion on the items of the list is present nor here nor in the manuscript.
The lack of a statistically significant tests in the field is the main drawback of the paper, since the results and the conclusion that “it is possible to design and implement the system for obstacle detection based only on two hydrophones in the BUV” (lines 397-398) is based on one test only, without any characterization in terms of errors “due to the impact of many factors on the accuracy” (lines 386-387).
In my opinion, before publication, the Authors have to show evidence of the reliability of the hydroacoustic system with multiple tests and with an evaluation of the errors and/or contingency tables showing the potential of their system in a real environment, with the acoustic noise that characterizes a real underwater ambient.
Please verify in the Bibliography the reference to papers [16] and [17], since the title of [17] seems to be the title of the paper present in the 18th International conference on Transport Science number 16, and the volume 67, number 1 of 2020 of the International Journal of Maritime & Technology seems to be non-existent.
In the following some minor comments:
Abstract: Lines 18 and 19. Which algorithms are you referring to?
Introduction: The introduction is confused: a real state of the art about the use of acoustic devices to avoid collision is missing. Just for example: part of the Introduction is related to the description of the past generation of BUV (lines 50-64), followed by a part on the near future research that will include also the communication between swarm of vehicles (lines 65-69) and some sentences about the Hilbert-Huang transform to estimate the range of multiple target (lines 70-78).
I recommend to rewrite the paragraphs of the Introduction with a more logical order.
Some misspell:
- line 72 “achieved” instead of “achieves”;
- line 86 “deals with” instead “deal with”;
- line 180 “as well as the window coefficient influence” instead “as well as and the window coefficients influences”;
- Delete lines 204-205 because they are a repetition of lines 173-174.
- Line 276 “pass-band” instead “pass-bad”
- Line 345: “Figures 8-11” instead “Figure 8-11”

Author Response

Thank you for your review of our paper. All your suggestions have been introduced into revised manuscript. We have answered each of your points below.

“The lack of a statistically significant tests in the field is the main drawback of the paper…

In my opinion, before publication, the Authors have to show evidence of the reliability of the hydroacoustic system with multiple tests and with an evaluation of the errors and/or contingency tables showing the potential of their system in a real environment, with the acoustic noise that characterizes a real underwater ambient.”

At the beginning we would like to explain the errors issue and number of tests. Although many test have been taken, only sixteen (each algorithm for eight measurements with four different courses 0 [deg], 30 [deg], 60 [deg], 90 [deg].) was depicted and discussed in the paper. This was mainly because many impact on the accuracy, what, in fact was written in lines 386-387.

Tests were done on the Wysockie lake (geographical coordinates: 54°24'44.0"N; 18°28'20.2"E), the Kosobudno lake (geographical coordinates: 53°50'06.7"N; 17°38'48.8"E) and the Gulf of Gdansk (Mechelinki, geographical coordinates: 54°36'36.7"N 18°30'51.1"E).

The data presented has been selected from a wide range of measurements.

Based on this measurements and after heated discussion, the decision was made don’t include chapter about the precision issue in this paper. There are too many factors influences on the hydroacoustic propagation and finally the system accuracy:

- the environmental factors (water temperature, water depth, type of bottom, distance from the shore, etc.),

- the measurements factors (type of propeller, depth of the BUV, etc.)

- devices parameters (kind of hydrophones, gain of the amplifier module, ADC resolution, etc.)

- algorithm used parameters (filter range, number of the measured data, etc.)

Only four course measurements are presented for the same state of the environment and one lake. The pontoon was not equipped with a specialized course mesurement, so the accuracy analysis would not be comprehensive.

Here, we focused on the difficult task of estimating the course of low speed propulsors (as in submarines) using only two hydrophones mounted at a closed distance. All stages of this analysis have been presented. What's more, the new method has been proposed and experimentally verified.

At that moment we are not able to provide sufficiently comprehensive error analysis.

We agree with you that accuracy analysis provides important information about the system and is always worth including in the technical documentation. But here, according to the authors experienced, there are too many factors to include them all in the article. Selected results will be published in a separate article.

In the paper it was mention that the test are going to be provided for different filter range and accuracy of bearing estimation is to be considered with keeping constant environment condition.

“Discussion session is just a collection of recommendation to the reader who wants to replicate and use the methods, but no discussion on the items of the list is present nor here nor in the manuscript.”

The presented system was the final version with a detailed description of the system used, type of hydrophones, sampling resolution, sampling frequency, filter used, filter parameters, etc. The recommendation has been prepared for those who would like to design a similar system. Some detailed information are included for example in text line 131-136 where is information about the risk of heap overwriting in one type of DSP. This recommendation helps to avoid difficult to identify heap overwrite problems. Other detailed information related to the type of hydrophones used is the gain of the amplifier (32 dB) and the 12-bit sampling resolution used. Suggestions 1-6 should be taken into account for followers. While number 7-8 are the proposition of used the new method if less computer power is demanding.

Taking under consideration the distance between hydrophones (1.25 m) and the sound wave length (75 m) no more than 1.5 % of samples are swift during the correlation calculation. This significantly reduce the memory and computational power demand (323-325 lines).

Which is crucial for an autonomous underwater vehicle.

“Please verify in the Bibliography the reference to papers [16] and [17], since the title of [17] seems to be the title of the paper present in the 18th International conference on Transport Science number 16, and the volume 67, number 1 of 2020 of the International Journal of Maritime & Technology seems to be non-existent.”

We apologise for that, but we did not noticed that the publication is on-line first it means it is not included in databased yet. I hope that this will be done soon. Below please find the information about the paper:

https://hrcak.srce.hr/index.php?show=clanak&id_clanak_jezik=338247&lang=en

“Abstract: Lines 18 and 19. Which algorithms are you referring to?”

In lines 18 and 19 we are referring to both algorithms. We apologise for the grammar mistake.

“Introduction: The introduction is confused: a real state of the art about the use of acoustic devices to avoid collision is missing. Just for example: part of the Introduction is related to the description of the past generation of BUV (lines 50-64), followed by a part on the near future research that will include also the communication between swarm of vehicles (lines 65-69) and some sentences about the Hilbert-Huang transform to estimate the range of multiple target (lines 70-78).”

The Introduction was rewritten according to the suggestion.

“Some misspell:

- line 72 “achieved” instead of “achieves”;

- line 86 “deals with” instead “deal with”…”

All the misspellings indicated by the Reviewer have been corrected. Moreover, the paper was verified by native speakers from the MDPI.

Round 2

Reviewer 2 Report

The paper was revised according to the remarques made. There are no further concerns from my side.

Author Response

Thank you very much for all your kind suggestion, which is valuable in improving the quality of the manuscript.
We appreciate the time and details provided by you and have incorporated the suggested changes into the manuscript to the best of our ability.
The manuscript has certainly benefited from these insightful revision suggestions.

Reviewer 3 Report

I appreciate the efforts made by Authors to improve the manuscript.

I understand that too many factors can impact on the hydroacoutic propagation to provide a comphrensive estimate on the accuracy of the system from a metrological point of view. On the other hand, the carried out experimental trials show that the developed algorithms are promising, but not the reliability of the system.

For this reason, before publication, I strongly suggest the Authors to:

1) modify the title in "Hydroacustic system to avoid collision with vessels with low-speed propellers in a controlled environment"

2) change lines 411-412 in "it is clear that bering estimation of sources of underwater sound, even at low frequency can be performed in a controlled environment".

Author Response

Thank you very much for all your kind suggestion, which is valuable in improving the quality of the manuscript.
We appreciate the time and details provided by you and have incorporated the suggested changes into the manuscript to the best of our ability.
The manuscript has certainly benefited from these insightful revision suggestions.
We hope that the revised version is now suitable for publication.